# Stress, Depression and/or Anxiety According to the Death by COVID-19 of a Family Member or Friend in Health Sciences Students in Latin America during the First Wave

Christian R. Mejia [1], Aldo Alvarez-Risco [2], Yaniré M. Mejía [3], Susan C. Quispe [3], Shyla Del-Aguila-Arcentales [4,*], Victor Serna-Alarcón [5,6], Martín A. Vilela-Estrada [6], Jose Armada [7] and Jaime A. Yáñez [8,9,*]

1   Translational Medicine Research Centre, Universidad Norbert Wiener, Lima 15073, Peru
2   Carrera de Negocios Internacionales, Facultad de Ciencias Empresariales y Económicas, Universidad de Lima, Lima 15023, Peru
3   Faculty of Health Sciences, Universidad Peruana de Ciencias Aplicadas—UPC, Lima 15023, Peru
4   Escuela de Posgrado, Universidad San Ignacio de Loyola, Lima 15024, Peru
5   Hospital José Cayetano Heredia, EsSalud, Piura 20002, Peru
6   Escuela Profesional de Medicina Humana, Universidad Privada Antenor Orrego, Trujillo 13001, Peru
7   Faculty of Business Sciences, Universidad Continental, Huancayo 12000, Peru
8   Facultad de Educación, Carrera de Educación y Gestión del Aprendizaje, Universidad Peruana de Ciencias Aplicadas, Lima 15023, Peru
9   Gerencia Corporativa de Asuntos Científicos y Regulatorios, Teoma Global, Lima 15073, Peru
*   Correspondence: sdelaguila@usil.edu.pe (S.D.-A.-A.); jaimeayanez@gmail.com (J.A.Y.)

**Abstract:** The COVID-19 pandemic generated high mortality in various countries, which may have had an impact on the mental health of young people. The objective of the study was to evaluate whether the death of a family member or close friend due to COVID-19 generated a higher prevalence of depression, anxiety, or moderate/severe stress in university health sciences students in Latin America. This is an analytical cross-sectional study, with secondary data; depression, anxiety, and stress were measured with a validated survey. In addition, data were obtained on the deaths by COVID-19 of family members or close friends, illness and other socio-economic variables. Descriptive and analytical statistics were obtained. It was found that, of the 3304 students, 5.9% (190) had a close relative who had died, 11.2% (363) a distant relative, and 19.8% (641) a friend. According to the multivariate analysis, those students who had a close family member who had died had greater depression (RPa: 1.48; CI 95%: 1.20–1.84; value $p < 0.001$) and stress (RPa: 1.41; CI 95%: 1.11–1.79; $p$ value = 0.005), in addition, those who had a friend who died had higher levels of anxiety (RPa: 1.20; 95% CI: 1.06–1.36; $p$ value =0.005); also, the respondents who suffered from COVID-19 had greater depression (RPa: 1.49; CI 95%: 1.05–2.11; value $p$ = 0.024) and stress (RPa: 1.55; CI 95%: 1.05–2.28, $p$-value = 0.028). An association was found between suffering from depression, anxiety, or stress, and having suffered the death of a family member or close friend from COVID-19. This finding is an important one for places of education to consider, suggesting a need to generate psychological support programs for students who have lost a loved one during the pandemic, since this could have academic and social repercussions. An association was found between the three mental illnesses studied and the death of a family member or close friend from COVID-19.

**Keywords:** depression; anxiety; stress; death; COVID-19; students

## 1. Introduction

Due to the COVID-19 pandemic [1], different measures were implemented, such as social distancing [2–4] and quarantines [5–8], to avoid contagion. However, the pandemic evidenced the weaknesses of the Peruvian healthcare system and public health policies [9]. Specifically, it made plain the lack of organizational support for workers in healthcare

facilities [10], generating a movement of workers to leave their jobs due to mental distress [11]. Populations, including young university students, were massively exposed to fake news and conspiracy theories [12], which generated technostress [13], many political controversies, and increased mental distress [14], generating the implementation of self-care behaviors [15], the search for preventive and curative measures with unproven drugs, and the self-use of medicinal plants, [16] in part based on their knowledge and appreciation of plants containing bioactive compounds [17–22]. Another significant impact was in terms of the loss of jobs in venues such as small firms [23,24], sports events [25], and the hospitality industry [26,27].

In this context, homes have become places of work, study, recreation, and leisure; thus, the need to change health self-care behaviors in the face of COVID-19 [15,28,29] was portrayed as an exercise in adapting to new lifestyles [30,31]. Various changes have been reported, such as the choice of eating behaviors in part based on their knowledge and appreciation of foods containing bioactive compounds [17–22,32–37], and adopting habits such as smoking, alcohol consumption, psychoactive substances, physical activity, and sex which, in turn, have implications for people's physical and psychological health [38–43].

Mental illnesses such as depression, anxiety, and stress are diseases that can afflict anyone regardless of race, sex or age. Likewise, they are conditions that have a high impact on public health, hence the importance of their being investigated [44]. According to the World Health Organization (WHO), "depression is a frequent mental illness, characterized by sadness, loss of interest or pleasure, feelings of guilt or lack of self-esteem, sleep or appetite disorders, feelings of tiredness and lack of concentration" [45]. Similarly, anxiety and stress are less severe disorders, but they alter the quality of life of people who suffer from them [46]. The WHO revealed that these mental illnesses affect more than 264 million people worldwide, and that this number is increasing [47].

Multiple studies reveal that university students are the leading group affected by these mental illnesses [13,48–50]. A study carried out at the San Antonia Catholic University of Murcia reveals one example of this, showing that health university students suffer at a higher prevalence from these illnesses [49]. In a university in Chile, 91% of the students who have these disorders were from the career-track in human medicine [50]. Likewise, a university in Lima-Peru study showed that medical students have a higher prevalence of depression (33.6%) compared to those studying other medical sciences and pursuing careers in health [51].

Within the current context, the situation experienced by the Coronavirus disease (COVID-19) produced by the SARS-CoV-2 virus brought with it a series of changes in different aspects of people's daily lives on a global level. COVID-19 began in November 2019 in the population center of Wuhan, China [52]; subsequently, the disease spread to almost all countries. After that, it was designated by the WHO as a pandemic [53]. COVID-19 mainly affects people with secondary risk factors, such as, among others, obesity and chronic diseases [54]. This characteristic of the SARS-CoV-2 virus led to a higher percentage of morbidity and mortality. Peru has been reported to be one of the countries in Latin America with the highest mortality rate [55]. The loss of millions of lives due to COVID-19 led many families to a state of mourning. In a study conducted at the San Cecilio Hospital in Granada-Spain, anxiety and clinical depression were found two months after the death of a relative in 30.3% and 21.1% of the participants, respectively; [56] one can appreciate in the study an association between the loss of a family member and levels of anxiety and depression, identifying it as a risk factor for suffering from these illnesses [57,58]. On the other hand, it was shown that the loss of a family member produces a series of neuropsychological changes such as alterations in the reward system, neurocognitive functioning, and neuronal systems involved in emotional regulation [56].

However, there are currently no studies in university health sciences that relate these mental conditions with the death of a family member and close friend, which was the subject of investigation in the present study; this could be relevant, since it was shown that the levels of anxiety, stress, and depression in university students of health sciences are

considerable [59–62]. A study carried out on university students of the medical school of Changzhi-China revealed that approximately 24.9% of the students surveyed experienced symptoms of anxiety [59]. In Italy, it was shown that 44% felt anxiety, and 48% experienced physical and psychological discomfort among students who did not receive psychological support [60]. In Latin America, various studies were carried out, which revealed an increase in stress, anxiety, and depression in students, such as the case of Mexico, where 31.9% of the population studied presented levels of stress and 40.3% of anxiety [61]. In Peru, a study conducted at the Ricardo Palma University revealed that 52.6% of students reported mild anxiety [62]. It is important to note that healthcare workers were the in the first line of defense during the first wave and the subsequent waves of the COVID-19 pandemic. It is also important to note that during the first wave of the pandemic there were not sufficient numbers of healthcare professionals to attend the high number of cases, and many healthcare students were requested to attend on an emergency basis or volunteered to do so [63–65]. Therefore, it is important to assess the stress, depression and anxiety of healthcare students caused by the death of a family member or friend during the COVID-19 pandemic first wave. Based on what has been reported, it has been shown that there are several risk factors related to increased stress, anxiety, and depression during the COVID-19 pandemic, which can be increased by the death of a relative or acquaintance suffering from SARS-CoV-2.

The objective of this study was to evaluate whether the death of a family member or close friend due to COVID-19 generated a higher prevalence of depression, anxiety and/or moderate or severe stress in health science students in Latin America during the pandemic. Furthermore, it will allow the researchers to assess the prevalence of these mental illnesses, the frequency in each country, the country that was most affected, the association between other socio-educational variables with the prevalence of these mental illnesses, and the proportion of students who lost close friends or relatives during the COVID-19 pandemic, taking into consideration that countries in Latin America have different lifestyles.

## 2. Methodology

### 2.1. Design and Place of Study

The study was an observational, analytical type that used a cross-sectional design. The study was done through secondary data analysis since the information had already been collected. The type of sampling used was convenience. The study population consisted of students of health sciences (medicine, dentistry, nutrition, and physical therapy) who were studying in the months of June-August in the different universities of Latin American countries (Peru, Chile, Paraguay, Mexico, Colombia, Bolivia, Panama, Ecuador, Costa Rica, El Salvador, Honduras, and Guatemala). These students come from public and private universities. These universities are in the urban areas of the above countries. The population of health science students was selected because, as evidenced by previous research, they were severely affected by mental illnesses because of the nature of their profession [66–68], the need for some of them to be in the front-line treating infected patients, [69–71] and the social stereotypes that appeared listing them as contagion sources [72–75].

### 2.2. Calculation of Statistical Power

In this project, the statistical power of each of the main intersections of the investigation was calculated, since, being a secondary data analysis, this would help us determine if the sample collected was sufficient for the analyses carried out. This measurement was made in the Stata 19 program, in which the independent variables were crossed with the exposure variables. This process was carried out at the time of data analysis, after the project's approval. It was found that only one cross had the power of less than 80%, this being whether you had a friend who died versus moderate or severe depression (57% power), and the other crosses obtained power of 100%. The power of the crosses was calculated according to whether you had a friend, a close relative or a distant relative who died from

COVID-19, this versus the three dependent variables (depression, anxiety and/or moderate stress).

### 2.3. Inclusion and Exclusion Criteria

The inclusion criteria were that the students be enrolled in the academic cycle during June, July, and August 2020, that they are health sciences students who agreed to participate in the research, that they are 18 years of age or older and that they resided in some Latin American country during the pandemic. Those who had errors in filling out the survey were excluded (excluding 189 surveys).

### 2.4. Study Variables

The dependent variables were suffering from stress, anxiety, and depression. Moderate and severe levels were taken as a category of interest, for which the DASS 21 [76] scale was used. This survey was completed by self-report and allowed the authors to evaluate the presence of symptoms of these illnesses. It is essential to mention that the survey has 21 items. Likewise, each question can obtain from 0 to 4 points. After adding the scores for each pathology, the cut-off points for moderate depression were from 19–25, for severe, a range of 21–27 was considered; moderate anxiety had a range of 10–14, severe anxiety extended from 15–19; and moderate stress went from 19–25 points, with severe stress at 26–36. The scale has been validated in multiple populations, one of the closest to our reality was the one carried out on university students in Chile, which obtained a Cronbach's alpha of 0.96 [77].

The primary exposure variables were the death of a family member and close friend, obtained through direct self-report questions. The other independent variables were:

a. The age of the participants;
b. Gender (female or male);
c. Country of residence (Peru, Chile, Paraguay, Mexico, Colombia, Bolivia, Panama, Ecuador, Costa Rica, El Salvador, Honduras, and Guatemala);
d. If they had a job or not, type of university (public or private);
e. If they suffered from COVID-19 (defined as the positive or negative diagnosis that the participant, a family member or friend had);
f. Current academic year;
g. If they had a domestic partner.

A close family member was defined as father, mother, husband, wife, son, daughter, brother, sister, grandfather, grandmother, father-in-law, mother-in-law, sister-in-law, brother-in-law, or domestic partner. A distant family member was defined as any other relative not in the list of close family members. A family member at home is defined as a relative that was living in the same house during the COVID-19 infection or death, while a family member away from home was at a different house.

### 2.5. Data Collection

This survey was conducted in Latin American countries during the relevant months with the main objective of determining the academic impact on undergraduate students in Latin America after the mandatory social isolation decreed in their countries due to the COVID-19 pandemic. The data collection was conducted virtually in the months of June, July, and August of the year 2020, through a questionnaire posted through the Google Forms format (https://forms.gle/Ra4YjRMmYS4XdWzM6, accessed on 15 September 2020), which is currently closed. For the present study, we had the collaboration of the Latin American Federation of Scientific Societies of Medical Students (FELSOCEM) in carrying out the data collection during the pandemic, and multiple respondents belonged to this federation.

*2.6. Quality Control*

Secondary quality control of the data from the obtained base was carried out (since the first process had already been carried out by the group that provided us with the information) through a double review process; this second review allowed us to ensure the quality of the information and, in this way, confirm compliance with the inclusion and exclusion criteria. Likewise, a dictionary of variables was created, which was later used to label the data in the statistical program.

*2.7. Ethical Aspects*

This study was based on ethical principles and considered the criteria of social value, scientific validity, and selection of the study population. This research work did not threaten the physical or emotional health of the respondents. Likewise, it complied with the criterion that they provide their consent, and that confidentiality is respected for each participant, since anonymity was maintained during the completion of the surveys, particularly respecting the right to identity privacy. The project was reviewed and approved by the Health Sciences Ethics Committee of the Peruvian University of Applied Sciences (UPC) (PI072-21), which verified that the protocol complied with the required ethical standards. Likewise, the database used in this project was reviewed and approved by the ethics committee of the Universidad Privada Antenor Orrego (Bioethics Committee Resolution No. 0239-2020-UPAO). Finally, this study had an essential social value to identify the risks of the topic raised.

*2.8. Data Analysis*

The data was processed in the Microsoft Excel program, for which this process was subjected to a double review, and the previously generated dictionary of variables was used. Subsequently, the data was entered into the Stata program (version 15). The univariate analysis of the categorical variables (death from COVID-19 of a family member or close friend, stress, depression, anxiety, sex, job, type of university, COVID-19, current academic year, and sentimental partner) was reported according to the frequencies and percentages. On the other hand, for the numerical variables (age and year of study), the median and interquartile ranges were found (this due to its non-normal behavior, was evaluated using the Shapiro Wilk statistical test). The initial bivariate analysis showed the association of the main variables; the Chi$^2$ test was used (according to the evaluation of the criterion of the minimum expected values). The strength of the association was found between the death of a relative or close friend from COVID-19 and the risk of suffering from stress, anxiety, and depression by obtaining the prevalence ratios (PR), the 95% confidence intervals, and the *p*-values. Further analysis was taken up by using the generalized linear models, with the Poisson family, the log link function, and with variance for robust models. For the elementary level and the adjusted model, a *p*-value <0.05 was considered significant (the primary criterion for a variable from the bivariate model to be considered for inclusion in the multivariate model).

**3. Results**

Of the total of 10,594 respondents, 7113 surveys were not included since these participants were not studying any career in health sciences. In addition, six surveys were excluded for not meeting the inclusion and exclusion criteria. Of the 3475 surveys, 183 were eliminated because they were invalid (Figure 1).

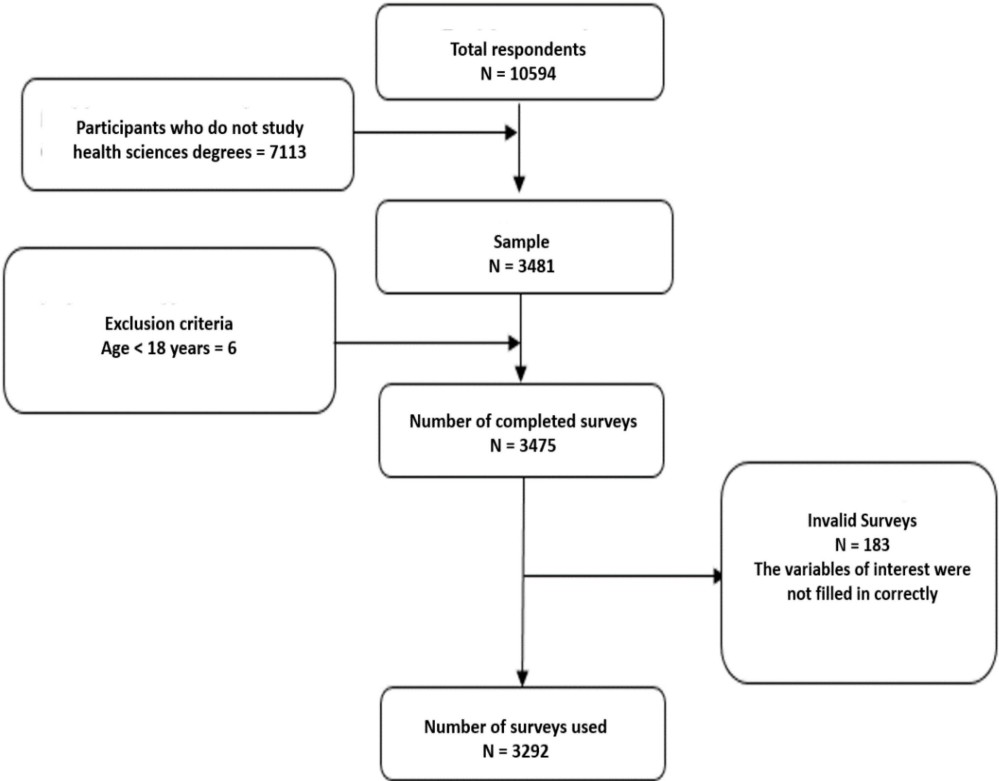

**Figure 1.** Flowchart of the selection of participants for the research work "Association of the presence of stress, depression and/or anxiety with death by COVID-19 of a family member or close friend in health sciences students in Latin America during the SARS-CoV-2 pandemic."

### 3.1. Socio-Educational Characteristics of University Students in 10 Latin American Countries

Of the 3292 students surveyed, 65.8% (2165) were women, and their median age was 20 years (interquartile range: 19–22 years), the median educational level was third-year (interquartile range: 1st–4th year of studies), 60.3% (1876) studied at a private university, 100.0% (3292) said they did not work, 65.2% (2147) did not have a partner, and 40.9% (1350) resided in Peru. In addition, 26.1% (811) suffered from moderate or severe depression, 31.5% (986) from anxiety, and 22.3% (689) from stress (Table 1).

**Table 1.** Socio-educational characteristics of university students in 10 Latin American countries.

| Variable | Frequency | Percentage |
|---|---|---|
| Sex | | |
| Female | 2165 | 65.8 |
| Male | 1127 | 34.2 |
| Age (years) * | 20 | 19–22 |
| Year of studies (years) * | 3 | 1–4 |
| Type of university | | |
| Public | 1234 | 39.7 |
| Private | 1876 | 60.3 |
| Work | | |
| No work | 3124 | 100.0 |
| Yes work | 0 | 0.0 |
| Sentimental couple | | |
| Single | 2147 | 65.2 |
| With couple | 1146 | 34.8 |
| Country | | |
| Peru | 1350 | 40.9 |
| Chile | 208 | 6.3 |
| Paraguay | 409 | 12.4 |
| Mexico | 315 | 9.6 |
| Colombia | 201 | 6.1 |
| Bolivia | 267 | 8.1 |
| Panama | 217 | 6.6 |
| Ecuador | 116 | 3.5 |
| Costa Rica | 90 | 2.7 |
| Honduras | 125 | 3.8 |
| Moderate or severe depression | | |
| No | 2292 | 73.9 |
| Yes | 811 | 26.1 |
| Moderate or severe anxiety | | |
| No | 2147 | 68.5 |
| Yes | 986 | 31.5 |
| Moderate or severe anxiety | | |
| No | 2404 | 77.7 |
| Yes | 689 | 22.3 |

* Median and interquartile range. Through the DASS-21 test.

### 3.2. Illness or Death from COVID-19 in the Immediate Environment of University Students in 10 Latin American Countries

As for deaths due to COVID-19, 5.9% (190) had a close relative who died, 11.2% (363) a distant relative, and 19.8% (641) a friend. 5.0% (162) had a relative at home with the pathology, 19.1% (618) had a relative outside their home, and 25.1% (815) a friend with the disease. Also, 1.8% (58) had suffered from COVID-19 at some point (Table 2).

**Table 2.** Illness or death from COVID-19 in the immediate environment of university students in 10 Latin American countries.

| Variable | Frequency | Percentage |
|---|---|---|
| Close family member died of COVID-19 | | |
| No | 3053 | 94.1 |
| Yes | 190 | 5.9 |
| Distant family member died of COVID-19 | | |
| No | 2880 | 88.8 |
| Yes | 363 | 11.2 |
| Friend passed away of COVID-19 | | |
| No | 2602 | 80.2 |
| Yes | 641 | 19.8 |
| Family member at home had COVID-19 | | |
| No | 3081 | 95.0 |
| Yes | 162 | 5.0 |
| Family member away from home had COVID-19 | | |
| No | 2625 | 80.9 |
| Yes | 618 | 19.1 |
| Friends had COVID-19 | | |
| No | 2428 | 74.9 |
| Yes | 815 | 25.1 |
| Suffered from COVID-19 | | |
| No | 3185 | 98.2 |
| Yes | 58 | 1.8 |

*3.3. Percentage of People Suffering from Anxiety, Depression and Stress-Related to Death from COVID-19 in the Immediate Environment of University Students in 10 Latin American Countries*

It was evidenced that a higher percentage of the total suffered from anxiety related to the death by COVID-19 of a friend (39.45%), a close relative (46.41%), and a distant relative (43.93%). Likewise, it was identified that the death of a member of the immediate environment due to COVID-19 generates a higher prevalence of suffering from stress, anxiety, and depression (Table 3).

**Table 3.** Percentage of people suffering from anxiety, depression and stress-related to death from COVID-19 in the immediate environment of university students in 10 Latin American countries.

| Variables | Depression N (%) | Anxiety N (%) | Stress N (%) |
|---|---|---|---|
| Friend died | | | |
| No | 631 (25.7%) | 732 (29.6%) | 525 (21.5%) |
| Yes | 171 (28.2%) | 243 (39.5%) | 157 (25.9%) |
| *p*-value | 0.22 | <0.001 | 0.019 |
| Close relative died | | | |
| No | 735 (25.5%) | 591(30.7%) | 623 (21.7%) |
| Yes | 67 (37.9%) | 84 (46.4%) | 59 (33.2%) |
| *p*-value | <0.001 | <0.001 | <0.001 |
| Distant relative died | | | |
| No | 701 (25.9%) | 823 (30.0%) | 581 (21.5%) |
| Yes | 101 (29.4%) | 152 (43.9%) | 101 (29.3%) |
| *p*-value | 0.158 | <0.001 | 0.001 |

Only the results of those who did have depression, anxiety, and moderate to severe stress are shown. The *p* values were obtained with the chi-square test.

### 3.4. Bivariate Analysis of Moderate/Severe Depression, Anxiety, and Stress According to Variables in University Students in 10 Latin American Countries

According to the bivariate analysis, correlation with severe depression was found with age ($p = 0.006$), the year of studies which they were in ($p < 0.001$), residence in countries such as Chile ($p = 0.001$), Panama ($p = 0.001$), and Honduras ($p < 0.001$), the death of a close relative ($p < 0.001$), the fact that there is a content of COVID-19 at home ($p = 0.032$) and that the respondent had become ill ($p = 0.004$). Regarding anxiety, associations were found with all variables except studying at a private university ($p = 0.066$), having a sentimental partner ($p = 0.092$) and residing in the country of Chile ($p = 0.076$). Stress was associated with sex ($p < 0.001$), age ($p < 0.001$), year of study ($p < 0.001$), having a romantic partner ($p = 0.036$), residing in Chile ($p < 0.001$), Panama (<0.001) and Honduras (<0.001), the death of a close relative (<0.001), distant relative ($p = 0.001$), friend ($p = 0.018$), sick relative at home ($p = 0.029$), sick relative away from home ($p = 0.002$), sick friend ($p < 0.001$), sickness of the respondent ($p = 0.049$) (Table 4).

**Table 4.** Bivariate analysis of moderate/severe depression, anxiety, and stress according to variables in university students in 10 Latin American countries.

| Variable | Moderate or Severe Mental Illness | | |
|---|---|---|---|
| | Depression | Anxiety | Stress |
| Sex | | | |
| Female | Reference category | Reference category | Reference category |
| Male | 0.89 (0.78–1.01) 0.068 | 0.89 (0.79–0.99) 0.041 | 0.63 (0.54–0.74) < 0.001 |
| Age (years) * | 0.96 (0.93–0.99) 0.006 | 0.95 (0.93–0.98) 0.001 | 0.93 (0.91–0.96) < 0.001 |
| Year of studies * | 0.87 (0.84–0.91) < 0.001 | 0.83 (0.80–0.87) < 0.001 | 0.89 (0.85–0.94) < 0.001 |
| Private university | | | |
| No | Reference category | Reference category | Reference category |
| Yes | 1.04 (0.92–1.17) 0.569 | 1.11 (0.99–1.24) 0.066 | 1.01 (0.88–1.16) 0.887 |
| With couple | | | |
| No | Reference category | Reference category | Reference category |
| Yes | 1.08 (0.95–1.22) 0.224 | 1.10 (0.99–1.22) 0.092 | 1.16 (1.01–1.32) 0.036 |

**Table 4.** *Cont.*

| Variable | Moderate or Severe Mental Illness | | |
|---|---|---|---|
| | Depression | Anxiety | Stress |
| Country | | | |
| Peru | Reference category | Reference category | Reference category |
| Chile | 1.38 (1.14–1.67) 0.001 | 1.16 (0.98–1.37) 0.076 | 2.10 (1.75–2.51) < 0.001 |
| Paraguay | 0.86 (0.70–1.04) 0.121 | 0.64 (0.53–0.77) < 0.001 | 1.01 (0.82–1.25) 0.893 |
| Mexico | 0.81 (0.65–1.01) 0.058 | 0.68 (0.56–0.83) < 0.001 | 0.96 (0.76–1.22) 0.727 |
| Colombia | 0.89 (0.69–1.15) 0.370 | 0.77 (0.62–0.97) 0.026 | 1.12 (0.86–1.46) 0.406 |
| Bolivia | 0.83 (0.65–1.05) 0.127 | 0.74 (0.60–0.91) 0.004 | 0.90 (0.69–1.18) 0.462 |
| Panama | 0.61 (0.45–0.83) 0.001 | 0.33 (0,23–0.47) < 0.001 | 0.46 (0.30–0.70) < 0.001 |
| Ecuador | 0.84 (0.60–1.18) 0.319 | 0.71 (0.52–0,96) 0.028 | 0.92 (0.62–1.35) 0.661 |
| Costa Rica | 1.11 (0.80–1.53) 0.548 | 0.64 (0.44–0.93) 0.020 | 1.13 (0.76–1.67) 0.556 |
| Honduras | 0.30 (0.17–0.55) < 0.001 | 0.28 (0.17–0.47) < 0.001 | 0.16 (0.06–0.42) < 0.001 |
| Close family member died | | | |
| No | Reference category | Reference category | Reference category |
| Yes | 1.48 (1.22–1.81) < 0.001 | 1.51 (1.28–1.79) < 0.001 | 1.53 (1.23–1.90) < 0.001 |
| Distant family member died | | | |
| No | Reference category | Reference category | Reference category |
| Yes | 1.14 (0.95–1.36) 0.151 | 1.46 (1.28–1.67) < 0.001 | 1.36 (1.14–1.63) 0.001 |
| Friend died | | | |
| No | Reference category | Reference category | Reference category |
| Yes | 1.10 (0.95–1.26) 0.215 | 1.33 (1.19–1.49) < 0.001 | 1.21 (1.03–1.41) 0.018 |
| Family sick at home | | | |
| No | Reference category | Reference category | Reference category |
| Yes | 1.29 (1.02–1.63) 0.032 | 1.47 (1.23–1.77) < 0.001 | 1.33 (1.03–1.71) 0.029 |
| Family sick away from home | | | |
| No | Reference category | Reference category | Reference category |
| Yes | 1.09 (0.94–1.26) 0.243 | 1.34 (1.19–1.51) < 0.001 | 1.27 (1.09–1.48) 0.002 |
| Friend sick | | | |
| No | Reference category | Reference category | Reference category |
| Yes | 1.10 (0.96–1.25) 0.164 | 1.30 (1.16–1.45) < 0.001 | 1.34 (1.16–1.54) < 0.001 |
| The respondent got sick | | | |
| No | Reference category | Reference category | Reference category |
| Yes | 1.64 (1.17–2.30) 0.004 | 1.49 (1.10–2.03) 0.011 | 1.50 (1.00–2.25) 0.049 |

Crude prevalence ratios (left), 95% confidence intervals (in parentheses), and *p*-values (right) were obtained with generalized linear models; with the Poisson family, the log link function and variance for robust models. * Variables are taken in their quantitative form. Depression, anxiety and stress were measured with the DASS-21 test.

*3.5. Multivariate Analysis of Factors Associated with Moderate/Severe Depression, Anxiety and Stress in University Students in 10 Latin American Countries*

According to the multivariate analysis, those students who had a close family member who died had greater depression (RPa: 1.48; CI 95%: 1.20–1.84; value *p* < 0.001) and stress (RPa: 1.41; CI 95%: 1.11–1.79; *p* value = 0.005), in addition, those who had a friend who died had higher levels of anxiety (RPa: 1.20; 95% CI: 1.06–1.36; *p* value =0.005); also, the

respondents who suffered from COVID-19 had greater depression (RPa: 1.49; CI 95%: 1.05–2.11; value *p* = 0.024) and stress (RPa: 1.55; CI 95%: 1.05–2.28, *p*-value = 0.028). The year of studies which they were in (*p* < 0.001), and residence in Chile (*p* < 0.001), Costa Rica (*p* = 0.035) and Honduras (*p* = 0.002) were also associated with suffering from depression. Anxiety was associated with being of female sex (*p* = 0.028), the year of studies (*p* < 0.001), residing in Chile (*p* = 0.005), in Mexico (*p* = 0.006), in Panama (*p* < 0.001) or in Honduras (*p* < 0.001) in comparison with Peru. Stress was associated with gender (*p* < 0.001), age (*p* = 0.006), year of studies (*p* = 0.043), having a romantic partner (*p* = 0.015), residing in Chile (*p* < 0.001), Paraguay (*p* = 0.003), Colombia (*p* = 0.026), Panama (*p* = 0.046), Costa Rica (*p* = 0.025) and Honduras (*p* = 0.002) (Table 5).

**Table 5.** Multivariate analysis of factors associated with moderate/severe depression, anxiety, and stress in 10 Latin American university students.

| Variable | Moderate or Severe Mental Illness | | |
| --- | --- | --- | --- |
| | **Depression** | **Anxiety** | **Stress** |
| Male | Did not enter the model | 0.88 (0.79–0.99) 0.028 | 0.64 (0.54–0.75) < 0.001 |
| Age (years) * | 0.99 (0.95–1.02) 0.395 | 1.01 (0.99–1.02) 0.512 | 0.95 (0.92–0.99) 0.006 |
| Year of studies * | 0.90 (0.85–0.95) < 0.001 | 0.88 (0.84–0.92) < 0.001 | 0.94 (0.89–0.99) 0.043 |
| Private university | Did not enter the model | Did not enter the model | Did not enter the model |
| With couple | Did not enter the model | Did not enter the model | 1.18 (1.03–1.36) 0.015 |
| Country | | | |
| Peru | Reference category | Reference category | Reference category |
| Chile | 1.52 (1.25–1.86) < 0.001 | 1.29 (1.08–1.54) 0.005 | 2.30 (1.89–2.80) < 0.001 |
| Paraguay | 1.09 (0.87–1.38) 0.453 | 0.89 (0.72–1.10) 0.290 | 1.48 (1.14–1.91) 0.003 |
| Mexico | 0.91 (0.72–1.14) 0.416 | 0.75 (0.61–0.92) 0.006 | 0.99 (0.77–1.27) 0.937 |
| Colombia | 1.12 (0.87–1.46) 0.380 | 0.97 (0.77–1.23) 0.823 | 1.37 (1.04–1.82) 0.026 |
| Bolivia | 1.05 (0.82–1.35) 0.679 | 0.94 (0.76–1.17) 0.568 | 1.13 (0.86–1.50) 0.380 |
| Panama | 0.79 (0.57–1.10) 0.160 | 0.44 (0.30–0.65) < 0.001 | 0.64 (0.41–0.99) 0.046 |
| Ecuador | 0.97 (0.69–1.39) 0.906 | 0.82 (0.60–1.12) 0.216 | 1.03 (0.69–1.54) 0.877 |
| Costa Rica | 1.44 (1.03–2.01) 0.035 | 0.89 (0.60–1.30) 0.538 | 1.59 (1.06–2.38) 0.025 |
| Honduras | 0.37 (0.19–0.70) 0.002 | 0.36 (0.21–0.61) < 0.001 | 0.21 (0.08–0.56) 0.002 |
| Close family member died | 1.48 (1.20–1.84) < 0.001 | 1.19 (0.99–1.43) 0.060 | 1.41 (1.11–1.79) 0.005 |
| Distant family member died | Did not enter the model | 1.14 (0.98–1.32) 0.090 | 1.15 (0.94–1.40) 0.182 |
| friend died | Did not enter the model | 1.20 (1.06–1.36) 0.005 | 1.18 (0.99–1.41) 0.058 |
| Family sick at home | 1.10 (0.85–1.42) 0.466 | 1.18 (0.97–1.44) 0.104 | 1.14 (0.87–1.49) 0.340 |
| Family sick away from home | Did not enter the model | 1.06 (0.93–1.21) 0.360 | 1.06 (0.89–1.26) 0.521 |
| Friend got sick | Did not enter the model | 1.09 (0.97–1.22) 0.167 | 1.14 (0.98–1.33) 0.091 |
| The respondent got sick | 1.49 (1.05–2.11) 0.024 | 1.25 (0.90–1.73) 0.181 | 1.55 (1.05–2.28) 0.028 |

Depression, anxiety and stress were measured with the DASS-21 test. Adjusted prevalence ratios (left), 95% confidence intervals (in parentheses), and *p*-values (right) were obtained with generalized linear models; with the Poisson family, the log link function and with variance for robust models. * Variables taken in their quantitative form.

## 4. Discussion

The death of a close family member was associated with higher levels of depression and stress among students, which may be caused by the fact that the loss of a family member produces drastic changes in the family environment and ways of living, which generate certain feelings of discomfort and emotional instability [78]. In the study carried

out by Romero and Cruzado, it was shown that 42% of the participants who suffered the loss of a close family member due to death presented depression (30%) and anxiety (21%) at clinically acceptable levels [56]. Likewise, a friend's death due to COVID-19 was associated with a higher level of anxiety. In a study conducted in the USA, an increase in mental health problems was identified due to the COVID-19 pandemic [79] with a prevalence rate after the death of a relative of 51% for anxiety and 48% for depression [80]. Likewise, it was determined that the risk of presenting these different mental illnesses would depend on multiple factors, mainly the affinity with the deceased person [81]. The unexpected loss of a family member and/or close friend is related to an increase in mental illness [82] and the severity of these illnesses will depend on multiple factors. In a study carried out in Australia, it was shown that the level of connection that is established or formed over the years plays a significant role in the presence of anxiety symptoms in the face of the loss of a friend [83]. However, suffering from stress, anxiety, and depression due to losing a friend is much lower risk than losing a close family member, which is influenced by other factors such as age, gender, race, religion, and interpersonal factors [83]. Where the respondent suffered from COVID-19, an association was found with higher levels of depression and stress. The very fact of suffering from a disease presupposes having to face a series of emotional reactions, as evidenced in different studies, since it is a predisposing factor to suffer from mental health problems [84–86]. Likewise, a study in patients diagnosed with COVID-19 at Huoshenshan Hospital (Wuhan-China) revealed that 35% and 28% of patients had symptoms of anxiety or depression, respectively [87].

Interestingly, the country of Chile had the three disorders at levels higher than those of Peru, even though Peru was at the time of the survey, in all the world, the country most affected by COVID-19 [56]. This may be very revealing data since, according to the WHO, Chile qualifies as one of the countries in the world with the highest morbidity due to psychiatric illnesses (at 23%), with major depression being the primary illness in the adult population [88]. Therefore, we can deduce that the increase in these figures was considerable and more significant than those of other countries, since they were already high before exposure to the experience of living through a pandemic (which led to having multiple limitations in daily life and to this is added the fact of losing a loved one to COVID-19). On the contrary, in Honduras, the three levels were obtained less frequently than in Peru. A study carried out in Honduras reports that measures focused on improving mental health had been established because stressors such as urban agglomeration, poverty, and inadequate working conditions prevailed in this population, which, according to the authors, are the causes of suffering from some mental illness [78]. In this way, it is hypothesized that those who reside in Chile are more affected by their mental health. On the other hand, those who live in Honduras already have levels of previous socio-political instability, which, in part, protects them from the events that they are currently living through [78]. These hypotheses could be verified in other investigations since they would generate different investigable scenarios for each reality.

Men had lower levels of anxiety and stress than women, which was evidenced in a study carried out at the University of Malaga, which indicates that women have higher anxiety levels due to certain psychosocial and biological factors [89]. What stands out most of these factors is vulnerability to exposure, since women of the present time are more vulnerable to exposure to stressful events [89]. The higher the age or year of studies, the lower the frequency of the three illnesses. It is known that the illnesses mentioned in our study are prevalent in the university population [90], which may be influenced by the beginning of a new stage and adaptation to university for first-year students, which may be stressful and a precipitating factor for suffering from anxiety, stress, and depression, and even more so, in the current circumstances, we face unfavorable situations [91]. Likewise, being in a higher degree of study is directly related to higher levels of stability and emotional maturity, which leads to being able to deal more adequately with various situations and seek professional help if necessary [92].

Finally, those who had a romantic partner presented more significant stress, which may be related to many factors involved in a relationship and the family environment in which the couple lives. It was evidenced based on research studies that an important risk factor for suffering from these psychiatric disorders is the loss of a loved one [93,94]. The sentimental partner is a support, assisting with the ability to cope with certain situations that may be risk factors for suffering from psychiatric disorders. The death of a family member and/or close friend is an event that affects the normal lifestyle of every person and even more so when adding the social isolation brought by the pandemic [95]. This can be related to many factors in which not only the fact of having a relationship is involved, but also the family environment in which the couple lives [95]. Likewise, the fact of the new biosecurity and confinement measures that were established to stop the wave of infections was perhaps one of the most challenging measures for people in this category, since they had to spend days without seeing the other in person or not being able to spend quality time, as they did before the pandemic. Therefore, when feeling the absence of their sentimental partner in daily life, it may be that the participants' stress levels rose. However, no research study has carried out this type of questioning. Therefore, based on evidence, it is impossible to affirm what the ultimate cause is, a question which could also be evaluated in future research specifically designed for this purpose.

This study presents results of mental health status of university students from various Latin American countries with different lifestyles, which gives us a more global and general vision. High levels of depression, anxiety and stress are to be expected because of the pandemic, but it is important to note that the population we selected (health science students), the time of the study (June to August, 2020, when there were no vaccines and the death toll in Latin America was high). Furthermore, this is a study with a high number of countries from this region, which makes it relevant and timely for a much larger audience.

*Limitations*

One of the limitations present in the study was due to its design (analytical cross-sectional) since, when measuring the exposure and the result at the same time, temporality cannot be seen, nor causality determined. However, associations were obtained, leading to the generation of hypotheses for those who carry out cause-effect investigations. Another limitation was that one of the crosses did not have the necessary power, being the cross of the variable that a friend died versus having moderate depression or more; therefore, this crossing must be taken with care for its analysis and interpretation. Likewise, there is the limitation of not being able to extrapolate the data to all the countries. However, the research tried to achieve a minimum sample size, as evidenced by adequate power in almost all crosses. Therefore, the results cannot be extrapolated to all the health sciences students in the surveyed countries. However, these results are significant as baseline or exploratory. Another limitation is the lack of a control group, such as students from other professions other than health sciences (medicine, dentistry, nutrition, and physical therapy) or the general population who are not students. Thus, it is not possible to extrapolate these results to all the university students in these countries or to the general population who are not students.

Since this study is a secondary analysis of a database, it can have certain biases in the sampling. For example, the bias derived from the use of the questionnaire: Because the data was collected from Latin America, the questions can be interpreted according to the sociocultural customs of each country. Furthermore, a possible memory error could also be present. The research group was exposed to memory bias, which is added to the fact of not being able to access other unmeasured variables (socioeconomic factors, access to internet services, family or personal history with a diagnosis of stress, anxiety, and depression, among others) that could be interesting due to the very fact that it was a secondary analysis of data. However, we obtained other essential variables, which could be considered as part of an initial study in multiple realities of a region.

From a practical perspective, it is important to consider these results when deciding whether to generate psychological support programs for students who lost a loved one during the pandemic, since this loss could have academic and social repercussions. In the current post-pandemic times and with the re-socialization that is occurring, post-traumatic stress disorders or other mental health effects are becoming more relevant since they might affect academic performance.

## 5. Conclusions

Finally, we conclude that an association was found between the death of a family member and close friend and the three mental illnesses studied (stress, anxiety, and depression). The results indicate a higher prevalence of these diseases than the rate of death by COVID-19. Likewise, it is important that, based on this study, the prevention and promotion of mental health in educational institutions be deepened. It is suggested that each institution carry out a situational analysis of these illnesses in their students to have an early diagnosis and then intervene by establishing the appropriate measures. In this way, they can seek optimal educational, personal and professional performance in their students.

**Author Contributions:** Conceptualization, C.R.M., A.A.-R. and J.A.Y.; Data curation, C.R.M., A.A.-R., Y.M.M., S.C.Q., S.D.-A.-A., V.S.-A., M.A.V.-E., J.A. and J.A.Y.; Formal analysis, C.R.M., A.A.-R., Y.M.M., S.C.Q., S.D.-A.-A., V.S.-A., M.A.V.-E., J.A. and J.A.Y.; Investigation, C.R.M., A.A.-R., Y.M.M., S.C.Q., S.D.-A.-A., V.S.-A., M.A.V.-E., J.A. and J.A.Y.; Methodology, C.R.M., A.A.-R., Y.M.M., S.C.Q., S.D.-A.-A., V.S.-A., M.A.V.-E., J.A. and J.A.Y.; Project administration, C.R.M., A.A.-R., J.A. and J.A.Y.; Resources, C.R.M., J.A. and J.A.Y.; Software, A.A.-R. and J.A.Y.; Supervision, C.R.M., A.A.-R. and J.A.; Validation, C.R.M., S.D.-A.-A. and J.A.Y.; Visualization, C.R.M., A.A.-R., S.D.-A.-A. and J.A.Y.; Writing—original draft, C.R.M., Y.M.M., S.C.Q., V.S.-A., M.A.V.-E. and J.A.; Writing—review & editing, C.R.M., A.A.-R., S.D.-A.-A. and J.A.Y. All authors have read and agreed to the published version of the manuscript.

**Funding:** The expenses required for data processing was funded by Universidad Norbert Wiener, project number 128-2022-R-UPNW.

**Institutional Review Board Statement:** The project was reviewed and approved by the Health Sciences Ethics Committee of the Peruvian University of Applied Sciences (UPC) (PI072-21), which verified that the protocol complied with the required ethical standards.

**Informed Consent Statement:** All the survey participants were well-versed in the study intentions and were required to consent before enrolling.

**Data Availability Statement:** The data presented in this study are available on request from the corresponding author.

**Conflicts of Interest:** The authors declare no conflict of interest.

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
