# Peer review of "Stress, Depression and/or Anxiety According to the Death by COVID-19 of a Family Member or Friend in Health Sciences Students in Latin America during the First Wave"

_sustainability, doi:10.3390/su142315515_

Round 1
Reviewer 1 Report
Dear authors, the work provides a very interesting and relevant approach at an international level and in multiple areas of human life.
His work is of high scientific quality. I am going to list some minor modifications that must be applied for its publication.
In the first place, they carry out an exhaustive definition of the sample, but do not include the sampling error. This has to be included.
Regarding the instrument, they must include the reliability and validity indices, as well as those obtained in this study. Similarly, the characteristics of the instrument must be described (number of items, dimensions it measures, psychometric properties, etc.).
Reference 76 does not have the same format as the rest. This has to be modified.
Reviewer 2 Report
Dear authors,
Thank you for submitting the manuscript: Stress, depression and/or anxiety according to the death by COVID-19 of a family member or friend in health sciences students in Latin America during the first wave. This study is interesting since it approaches such an important aspect as mental health and food consumption in the pandemic period, we are living in. I suggest accept it in present form.
Best regards.
Reviewer 3 Report
The authors brought up the timely topic and analyzed the data well. However, in my review, I found that the purpose of the study was not clear, and the findings and discussion were not related to the main topic.
Introduction:
· Some contents are not relevant to the topic.
· The purpose of the study is too general (not clear which issues were addressed)
· Explain why health sciences students were selected for this study
· In-depth theoretical backgrounds needed.
Discussion:
· There is a gap between the aims of the study and the discussion.
· Your findings just indicated that the health sciences students’ stress, depression, and anxiety are high, which is expected of every individual who experienced a family or close friend’s death due to COVID-19.
· Need more explanation on why the health sciences students were selected and what is the meaning of the findings without a comparison group.
· There is an inconsistency between the purpose of the study, findings, and discussion.
Round 2
Reviewer 3 Report
I am sure this topic is critical to be addressed and a timely study. Thank you for the revision, but I still see some areas that are not clear. I hope my comments will help the authors improve the manuscript.
1. Introduction
· Psychological disorders such as depression, anxiety, and stress are diseases that can afflict anyone regardless of race, sex or age.
o The following languages (Psychological disorders, diseases, vs. mental health vs. mental illness) should be carefully and consistently used.
· It is important to note that healthcare workers were in the first line of defense during the first wave and the subsequent waves of the COVID-19 pandemic. It is important to note that during the first wave of the pandemic there were no sufficient healthcare professionals to attend to the high number of cases. Multiple healthcare students were requested to attend the sanitary emergency or volunteered to do so [63-65]. Therefore, it is important to assess the stress, depression, and anxiety of healthcare students caused by the death of a family member or friend during the COVID-19 pandemic's first wave.
o Still there is no clear purpose of the study.
o Provide statistics about the proportion of students that lost close friends or relatives during the COVID-19 pandemic
They may be impacted by patients’ death or patients’ family members during
the pandemic, instead of their own family or relatives. Provide more evidence with in-text citations.
2.4 Study variables
· This survey was completed by self-report and allowed evaluate the presence of symptoms of these pathologies.
o I wonder if the language ‘pathologies’ is appropriate.
· There is no question in the survey asking about the death of a family member or close Friend.
Table 3: clarify the total of participants (n=?) for each condition.
4. Discussion
· Regarding the fact that the respondent suffered from COVID-19, an association was found with higher levels of depression (maybe anxiety) and stress in this group of people.
-Indicate a table supporting your argument.
· For further study, the hypotheses do make sense in terms of mental health conditions, but not related to the present study.
· Discussion should be closely related to your research questions and hypotheses.
· I want to see more clear research questions, and an in-depth literature review relevant to the topic (the relationship between conditions of depression, anxiety, and stress and loss of a family or close relatives during the pandemic) instead of general information about depression, anxiety, and stress.
² Overall, it is a timely topic using a good database, but there are no clear research questions and no consistency between the sections.
Round 3
Reviewer 3 Report
Thank you for the authors' hard work on the revision.